## [Peer Review File · Nature Communications]

Reviewers' comments:

Reviewer #2 (Remarks to the Author):

The authors have responded positively to the reviewer comments and I believe the manuscript is improved as a result. The addition of the ICC/reliability of the modelling and the replication dataset are key improvements. Despite these improvements however, there are still a few things that I think could be clarified, particularly around the performance of the CNN model relative to other approaches and the real-world relevance of the genetic effects. Specifically:

I'm still not really convinced that the performance of their CNN approach has been properly compared with existing approaches. SVR is employed separately three times using three different types of input features, the lowest MAE = 4.37 (for VBM). When the CNN model uses single types of input features the lowest MAE = 4.01. This is some improvement, though only moderate. In fact, the use of the CNN with GM images (which is arguably the closest to the SVR VBM) has MAE = 4.64, so performs slightly worse. Really, my point here is that the comparison has not been done directly. The SBM, VBM and SM inputs could have been used in the CNN context, and more importantly the performance improvements of the CNN method are only really apparent when combining input feature types (e.g., T1 + JM). Could SVR not have been run using combined input feature types (e.g., using majority voting)? And why did they pick SVR? Gaussian Processes or Relevance Vector regression have been much more commonly used in the context of brain-age prediction.

Regarding the genetic effect size, it seems I made a miscalculation and that the 'age' effect size of the top SNP is in fact -0.32 years instead of -0.08 years. Perhaps the fact of the unit normalisation of the PAD values can be made clearer. Moreover, I think the authors still need to discuss the meaningfulness of the SNP influencing someone's brain age by approximately 117 days. For example, putting this effect in the context of the magnitude of aberrant brain ageing observed in studies of neurological or psychiatric diseases (which are generally much larger). I think it's essential in GWAS and similar research to show an awareness of the difference between statistical significance and real-world relevance.

The findings in the replication sample of the UKB are critical and should be up-weighted and the supplementary table (Table 12 Appendix G) included in the main text. If number of items are a problem, then these results should be incorporated into Table 4.

I'm still not convinced that gender and scanner should be included in the model, as the authors themselves show neither relate to age (as would be expected). I don't really expect the authors to re-run everything without them, but I think some mention of this as a potential red herring should be included in the Discussion. After all, people may want to replicate the analysis in future, and in such circumstances, I don't think gender and scanner should be used in this way.

Pearson's correlations are reported as % throughout. My understanding was that Pearson's r was always between -1 and 1. This should be clarified.

Did the authors include the extra genetic QC information requested by Reviewer 3? Not clear from the rebuttal letter. This level of methodological rigour is important.

What software did the authors use to code their CNN? Can they stick their model code up on Github or similar please?

Reviewer #3 (Remarks to the Author):

NCOMMS-19-09419-T

Deep learning based brain age prediction uncovers associated sequence variants

This paper by Jonssona et al. has been improved. However, some of my major concerns haven't been fully addressed,

and I think there is still much room to improve the transparency and clarity of the manuscript.

Detailed comments:

* I am disappointed that the authors still did not remove related individuals and properly adjust for nuisance variables such as head motion, genotyping array, imaging center, and even PCs, in their GWAS.

This is not about whether these covariates have large effects on results/conclusions, it's about the quality, rigor and integrity of science.

In addition, adjusting for PCs doesn't adequately account for the potential bias induced by relatedness.

Therefore, I urge the authors again to follow the best practice of GWAS, removing one individual from each related pairs (relatedness estimates have been released by UK Biobank), and adjust for all potential nuisance variables (head motion, ICV, genotyping array, imaging center, PCs, age at the imaging visit, sex, and perhaps age^2 , $age \times sex$, $age^2 \times sex$, plus other relevant covariates)

in one regression model and update the results in the manuscript.

Similarly, covariates should be properly adjusted when testing the associations between PAD and cognitive measures.

Separate, posthoc assessments of the effect of each of these covariates on SNP associations are not acceptable to me.

* The authors showed that by a transfer learning in IXI, the performance of CNNs trained in the Icelandic sample significantly improves in the UK Biobank.

I was curious what if the transfer learning was performed in a small subset of UK Biobank sample? The prediction is supposed to be more accurate, right? If so, shouldn't the authors use this more accurate prediction for GWAS discovery?

* Split of datasets into training, validation and testing seems to be arbitrary. Is it possible to randomly split the datasets say 100 times and assess the variability of prediction?

* Page 15: please report the SBM and BVM features used in the model in the supplementary.

* Page 7: "we did not spend as much time optimizing these CNNs"; Page 9: "the new PAD estimates are less optimized".

What does less optimized exactly mean? Why not use a consistent procedure, same as what was used in the original PAD estimation?

* Table 2: why test R^2 can be negative?

* Table 3 only presents selective results rather than all cognitive tests.

If no association was found for other tests, it should be mentioned in the main text. Detailed statistics should be reported in supplementary.

* LocusZoom plot should be presented for the two GWAS significant SNPs. In particular, the one on chr1 doesn't seem to be very reliable.

It's not GWAS significant with random CNN weight initialization.

And remember that some covariates haven't been properly controlled.

* The authors mentioned that PAD was estimated to be heritable in prior studies. It would be helpful to compute the heritability of PAD in the UK Biobank as a comparison.

* Computational cost (time and memory use) of CNN should be reported.

* The flow of the text can be further improved. Currently one needs to go back and forth between the results and methods to figure out what has been done.

* Page 9: "We also see that the other three sequence variants and PAD are associated with numerous structural brain phenotypes"

This is sort of circular because search of the three sequence variants were restricted to those associated with brain phenotypes.

* Page 9: "we scanned for the phenotype effects of all five single-nucleotide polymorphisms (SNPs) in the UK Biobank data analyzed by the Roslin Institute"

The main text should be more self-contained even if detailed results are presented in the supplementary.

What are those phenotypes? Is this a PheWAS study? What is the main finding/conclusion?

* Page 11: "the assessment center where neuropsychological testing was performed"  "the assessment center where neuropsychological testing was performed"

We want to thank the *Nature Communications* referees for their insightful comments. We have gone thoroughly through their reviews and answered them point-by-point (see text below) and revised the manuscript accordingly. The key points we have addressed in the new version of the manuscript are concerns about the comparison of our method with existing approaches and potential confounds in the GWAS and cognitive assessment association test. We have added results for regression methods that have previously been effective in brain age prediction (relevance vector regression and Gaussian Process regression) and combined models trained on SBM, VBM, SM features (Section 2.1). We found that our combined CNN method still had lower test MAE than these newly added methods. The GWAS and cognitive assessment association tests were re-run after removing individuals from related pairs and adjusting the PAD for various potential nuisance variables. Numerous items in the manuscript were updated with these new results (Tables 3, 4, 6, and 13 and Figures 2, 13, and 14). However, when comparing these updated results to the original result, we found that the additional adjustments had a small effect. Additionally, we have added SNP-heritability estimates for PAD (Section 2.5), LocusZoom plots showing the GWAS significant variants (Figure 13 and 14), information about computational cost (Section 4.3), association results for all of the cognitive tests considered (Table 6), and tables including the names of the SBM and VBM features used in our models (Supplementary table 1 and 2).

1 Reviewer 2

I'm still not really convinced that the performance of their CNN approach has been properly compared with existing approaches. SVR is employed separately three times using three different types of input features, the lowest MAE = 4.37 (for VBM). When the CNN model uses single types of input features the lowest MAE = 4.01. This is some improvement, though only moderate. In fact, the use of the CNN with GM images (which is arguably the closest to the SVR VBM) has MAE = 4.64, so performs slightly worse. Really, my point here is that the comparison has not been done directly.

- a. These points are mostly fair, however, we do not agree that the CNN with GM inputs is the closest to SVR VBM since the latter includes information about GM, WM and CSF (see supplementary table 2 for VBM variables). Therefore, a closer match to the SVR VBM method should be the CNN trained on T1 images. This method has test MAE = 4.01, which is lower than the test error of the SVR VBM method (MAE = 4.37).

The SBM, VBM and SM inputs could have been used in the CNN context, and more importantly the performance improvements of the CNN method are only really apparent when combining input feature types (e.g., T1 + JM). Could SVR not have been run using combined input feature types (e.g., using majority voting)?

- a. This is a true, we could have combined the predictions based on SBM, VBM, and SM inputs like we did for the CNNs. We have added results to Table 1C for regression models trained on SBM, VBM, and SM inputs and combined using both majority voting and linear regression blending. Additionally, we have added to Section 4.4 details about how these combinations were performed:

"Additionally, we tested combining predictions made by models trained on these three feature types. We decided to pick predictions for the method with the lowest CV MAE¹ for each feature type. Specifically, these regression methods were GPR with a Matérn kernel for both the SBM and VBM, and ridge regression for the similarity matrix features. Since the SBM and VBM features were not available for every image in the Icelandic dataset, we first calculated the average brain age prediction for each subject, before inner joining the predictions by subject into a single data frame. This resulted in a combined data frame with 1246 rows and three columns containing brain age predictions for the three regression methods. Since training these regression models is faster than training the CNNs, we were able to combine them using 10-fold cross validation predictions. Thus, the linear regression blender can train on predictions from the whole training set, instead of being limited to the 298 images in the validation set, as is the case for the CNN prediction combination."

Regarding the accuracy, we found that combining predictions using linear regression blending

¹The methods were picked by CV MAE instead of test MAE to prevent data leakage.

results in a test MAE = 3.849 years. While these predictions are more accurate than any of the individual SBM, VBM, and SM input predictions, they are less accurate than the predictions made by the CNNs combined with linear regression blending [MV (T1, JM, GM, and WM)] (MAE = 3.388 years). Similarly, the CNNs combined using majority voting have lower test error (MAE = 3.584 years) than the regression models trained on SBM, VBM and SM inputs and combined using majority voting (MAE = 3.940 years).

And why did they pick SVR? Gaussian Processes or Relevance Vector regression have been much more commonly used in the context of brain-age prediction.

- a. To be clear, we did not only look at SVR. We also looked at linear regression, lasso, ridge regression, elastic net regression, and random forest regression. Table 1C only displays the results for the method that has the lowest test error. The list of methods we tried was of course far from exhaustive, instead these methods were chosen to represent commonly used and relatively simple to tune regression methods.

In regard to Gaussian processes regression (GPR) and relevance vector regression (RVR), the main reason why we originally only included SVR but not GPR and RVR was because all three of these methods are kernel regression methods [1] and we did not expect to see a large difference in accuracy between them. However, as the reviewer mentions, it is true that RVR and GPR have previously been successfully used for brain age prediction [5, 7]². Taking the reviewer's point into consideration, we decided to add prediction results for both RVR and GPR for the SBM, VBM, and SM features to Table 5 in Appendix B. Additionally, we found that GPR trained on VBM features had a slightly lower test MAE (4.317 years) than the previous best SVR model (4.368 years). Therefore, the VBM feature results in Table 1C were replaced with the GPR prediction results.

Regarding the genetic effect size, it seems I made a miscalculation and that the 'age' effect size of the top SNP is in fact -0.32 years instead of -0.08 years. Perhaps the fact of the unit normalisation of the PAD values can be made clearer. Moreover, I think the authors still need to discuss the meaningfulness of the SNP influencing someone's brain age by approximately 117 days. For example, putting this effect in the context of the magnitude of aberrant brain ageing observed in studies of neurological or psychiatric diseases (which are generally much larger). I think it's essential in GWAS and similar research to show an awareness of the difference between statistical significance and real-world relevance.

- a. The reviewer questions the real-world relevance of the association of a single SNP with PAD. The reviewer is absolutely right that statistical significance does not ensure relevance. This is true in most fields of science, particularly when multiple testing and large samples fuel discovery.

However, comparing the effect of a single common SNP on the prediction of brain age in a relatively healthy sample with the overall effect of certain (severe) neurological diseases on brain aging, is not necessarily the most fruitful approach. First, geneticists do not expect any single common SNP to account for a large fraction of the trait variance, as this rarely happens. Second, while the direction of a significant association is clear, the effect size for each SNP is simply not well determined. Thirdly, the association may be indirect, due to linkage disequilibrium with the actual culprit. Thus, the GWAS field mostly emphasizes significance and direction of the observed effects rather than the effect sizes. Hence, the comparison suggested by the reviewer is not really meaningful, i.e. comparing the observed effect size for a single SNP to the overall effect of a (severe) neurological or psychiatric disease, representing the combined effects of all biological and environmental factors (including medication).

The notion that GWAS findings are somehow overvalued, representing "mere associations" devoid of causal inference, does not always recognize that this is very often the situation using other lines of scientific enquiry. Furthermore, polygenic scores, based on GWAS results for a large number of common SNPs weighted by their effect sizes, have shown utility and predictive power, but have also been criticized as they initially explained a small part of the variance and heritability of the traits studied. This is changing, and a recent manuscript, available on BioRxiv, reports that using

²However, we would like to point out that SVR was also used in Franke et al. [7]. Their results show that the accuracy of the best SVR model was only slightly worse than the accuracy of the best RVR model.

both common and rare variants from GWAS, the heritability of both BMI and adult human height can be entirely recovered (Wainschtein et al. [11]). Hence, for these traits at least, GWAS results can fully capture the biological basis as represented by other measures of heritability. Regarding GWAS on brain ageing, it is still early days. Our findings from the application GWAS to PAD, the identification of associations with a number of SNPs with PAD, provide proof of principle (the GWAS yields a handful of associated sequence variants) and a foothold for further studies. In response to these concerns we have added the following:

We have added a footnote to Table 4 explaining how to estimate the effect size in years:

"The reported effect sizes in Table 4 are for PAD normalized to unit variance. Before normalization the standard deviation of PAD was ~4 years. Thus the associated lowering of the protective allele of rs1452628 is approximately -0.32 years."

In the discussion we have made the following modification:

"Applying our method to large genomic datasets such as the UK Biobank has enabled us to identify novel associations between sequence variants and brain ageing. The variants identified are common SNPs with small effects on PAD (Table 4) accounting for only a fraction of the trait variance. However, these first findings provide a foothold, and further research into these loci as well as additional GWAS studies have potential to shed light on the biological underpinnings of the ageing brain and its connection to various diseases and disorders."

The findings in the replication sample of the UKB are critical and should be up-weighted and the supplementary table (Table 12 Appendix G) included in the main text. If number of items are a problem, then these results should be incorporated into Table 4.

- a. This is a good idea, we have followed the reviewers suggestion and incorporated the replication results into Table 4.

I'm still not convinced that gender and scanner should be included in the model, as the authors themselves show neither relate to age (as would be expected). I don't really expect the authors to re-run everything without them, but I think some mention of this as a potential red herring should be included in the Discussion. After all, people may want to replicate the analysis in future, and in such circumstances, I don't think gender and scanner should be used in this way.

- a. It is corrected that we previously showed that age does not associate with sex and scanner. However, we do not see how including these variables in the brain age prediction model should impact the replicability of our analysis. Sex is known to influence brain structure [10] and acquisition protocols are known to affect brain volume estimates [9]. The input features to our brain age prediction model are brain structure measurements and the goal of these models should be to accurately detect changes in brain structure that are truly caused by ageing. This is especially important when attempting to derive biologically relevant markers such as PAD. It seems intuitive that if we include sex and scanner in the model it can learn to disentangle and ignore changes in brain volume due to these factors faster than if it has no information about them. Therefore, it is not clear to us how using these variables could be construed as a potential red herring.

Pearson's correlations are reported as % throughout. My understanding was that Pearson's r was always between -1 and 1. This should be clarified.

- a. Yes, Pearson's r is always between -1 and 1. We have changed the format of all Pearson's correlation results from percentages to the more standard decimal fraction format to prevent distracting the reader.

Did the authors include the extra genetic QC information requested by Reviewer 3? Not clear from the rebuttal letter. This level of methodological rigour is important.

- a. Information about the genetic QC procedures that were performed is located in Section 4.5. For more information see the reply to Reviewer 3 below.

What software did the authors use to code their CNN?

- a. We used Keras with TensorFlow backend to code the CNNs. A footnote has been added to Section 4.3 with this information:

"The deep learning models were implemented using Keras with TensorFlow backend [4]."

Can they stick their model code up on Github or similar please?

- a. We will add the code to GitHub when the paper is published.

2 Reviewer 3

I am disappointed that the authors still did not remove related individuals and properly adjust for nuisance variables such as head motion, genotyping array, imaging center, and even PCs, in their GWAS. This is not about whether these covariates have large effects on results/conclusions, it's about the quality, rigor and integrity of science. In addition, adjusting for PCs doesn't adequately account for the potential bias induced by relatedness. Therefore, I urge the authors again to follow the best practice of GWAS, removing one individual from each related pairs (relatedness estimates have been released by UK Biobank), and adjust for all potential nuisance variables (head motion, ICV, genotyping array, imaging center, PCs, age at the imaging visit, sex, and perhaps age^2 , $age \times sex$, $age^2 \times sex$, plus other relevant covariates) in one regression model and update the results in the manuscript. Similarly, covariates should be properly adjusted when testing the associations between PAD and cognitive measures. Separate, posthoc assessments of the effect of each of these covariates on SNP associations are not acceptable to me.

- a. We are of course dedicated in preserving scientific integrity and want to follow the best practice of GWAS. As such, these points have been taken into consideration in the new version of the manuscript. Individuals from related pairs have been removed and PAD was adjusted for these potential nuisance variables before re-running the GWAS and cognitive measure association tests. The specific items that have been updated in the manuscript are Tables 3, 4, 6, and 13 and Figures 2, 13, and 14. However, when we compared these updated results to the original result, we found that non of the additional adjustments significantly altered the results. We have added the following description of the performed adjustments to the methods section:

In Section 4.5 we have updated the description of the adjustments performed on PAD before GWAS:

"Before performing GWAS, we removed individuals of non-white British ancestry and subjects from related pairs. We then adjusted the PAD for age at imaging visit, age^2 , sex, $age \times sex$, $age^2 \times sex$, total intracranial volume, the first 40 principal components from genetic ancestry analysis, head motion, genotyping array, and imaging center using linear regression. The adjusted PAD was then normalized with an inverse normal transformation. After normalization the linear regression adjustments were reapplied."

Additionally, we have updated the description in Section 4.5 for the adjustments performed on PAD before testing for association between PAD and cognitive measures:

"Before performing the association test we first removed individuals of non-white British ancestry and subjects from related pairs. We then adjusted the PAD for age at imaging visit, age^2 , sex, $age \times sex$, $age^2 \times sex$, total intracranial volume, the first 40 principal components from genetic ancestry analysis, head motion, genotyping array, imaging center, and assessment center where neuropsychological tests were conducted. The adjustments was performed using linear regression."

The authors showed that by a transfer learning in IXI, the performance of CNNs trained in the Icelandic sample significantly improves in the UK Biobank. I was curious what if the transfer learning was performed in a small subset of UK Biobank sample? The prediction is supposed to be more accurate, right? If so, shouldn't the authors use this more accurate prediction for GWAS discovery?

- a. While it is likely that transfer learning on a small subset of the UK Biobank sample will lower the UK Biobank test MAE. We refrained from doing this because we want to use as many subjects as possible in the downstream analysis and because of the limited age range in the UK Biobank sample (all subjects are in the age range 45 to 80 years). Training on such a limited age range would severely bias the model towards predicting ages inside this range. To get around this, it is necessary to train the model on a sample with a wider age range. This is why we transfer learned on the IXI sample, which includes subjects in the age range 20 to 86 years, before carrying out the downstream analysis.

We have added a paragraph to the end of Section 2.2 to clarify this:

"In subsequent sections, the CNNs trained with transfer learning on the IXI sample will be used in downstream analysis of the UK Biobank sample. While it is likely that transfer learning on a small subset of the UK Biobank sample will lower the UK Biobank test MAE, we refrained from doing this because we want to use as many subjects as possible in the downstream analysis and because of the limited age range in the UK Biobank sample (all subjects are in the age range 45 to 80 years). Training on such a limited age range would severely bias the model towards predicting ages inside this range. To get around this, it is necessary to train the model on a sample with a wider age range. This is why we use the CNNs trained on the IXI sample, which includes subjects in the age range 20 to 86 years, in the downstream analysis."

Split of datasets into training, validation and testing seems to be arbitrary. Is it possible to randomly split the datasets say 100 times and assess the variability of prediction?

- a. It is possible to train our brain age prediction 100 times. However, this would be extremely time consuming using the resources available to us. It would entail training four CNNs 100 times. Using four parallel GPUs, we expect the training to take at the very least 100 days. We already trained the model four additional times (Section 2.3) and did not see much variability in the PAD (ICC=0.86). We could train more models but we do not expect the results to change much.

Page 15: please report the SBM and BVM features used in the model in the supplementary.

- a. The names of the 311 SBM and 204 VBM features used to train the brain age prediction models have been added to Supplementary table 1 and 2.

Page 7: "we did not spend as much time optimizing these CNNs"; Page 9: "the new PAD estimates are less optimized".

What does less optimized exactly mean? Why not use a consistent procedure, same as what was used in the original PAD estimation?

- a. A consistent procedure was used when training the CNNs for the new PADs. However, it is well known that the training of deep learning networks is equivalent to the minimization of a non-convex cost function. This means that 1) the cost function can have local minima and saddle points that the optimization algorithm can get stuck in, and 2) different initialization can lead to different solutions. In view of this it makes sense to try different random initialization points and focus on the network that corresponds to the best local minimum, i.e., the CNN that has the lowest validation error. This was carried out when we trained the original method but could not be done when we retrained the CNNs in Section 2.3. Even though the retrained CNNs did not find the exact same solution as the original CNN, we found that the agreement between the original PAD and the four new PADs calculated with the retrained CNNs is quite high (ICC=0.86).

We changed the sentence on page 7 to clarify this:

"The reason why the error is higher here compared to the original results is that we did not reinitialize and retrain the CNNs in cases where the optimization got stuck in a poor local minimum or a saddle point."

The sentence on page 9 has been removed in response to a comment below (see the reply).

Table 2: why test R^2 can be negative?

- a. There are several definitions of R^2 , we used the scikit-learn implementation³ of R^2 . The possibility of negative R^2 values is noted in the scikit-learn documentation:

"Unlike most other scores, R^2 score may be negative (it need not actually be the square of a quantity R)."

Specifically, the R^2 is negative when the model fits worse than a horizontal line⁴.

Table 3 only presents selective results rather than all cognitive tests. If no association was found for other tests, it should be mentioned in the main text. Detailed statistics should be reported in supplementary.

- a. All association results for the cognitive tests we performed have been added to Table 6 (Appendix C). Additionally, we have added a sentence to Section 2.4 clarifying which cognitive tests did not show evidence of association with PAD:

"We did not find evidence of association between PAD and performance on the fluid intelligence, numeric memory, pairs matching, and prospective memory tests."

LocusZoom plot should be presented for the two GWAS significant SNPs. In particular, the one on chr1 doesn't seem to be very reliable. It's not GWAS significant with random CNN weight initialization. And remember that some covariants haven't been properly controlled.

- a. As per the reviewers request, LocusZoom plots for the two GWAS significant SNPs have been added to the manuscript (Figure 13 and 14 in Appendix I). We plotted the UK Biobank PAD GWAS results for a 0.2 Mb radius around the two GWAS significant variants (Figures 13 and 14). The LD information was estimated from the European population in the 1000G phase 3 release from Nov 2014.

The chr1 variant is not unreliable, it is associated with PAD in the replication set (effect = -0.07, $p=8.8e-04$) and associated with increased CSF in multiple cortical regions which is consistent with increased brain age (Table 8). The reason for the apparent discrepancy is because the random weight initialization CNNs converged to less optimal solutions (see the above discussion) than the brain age prediction CNN used in the GWAS. Which in turn results in lower association values for the PAD estimated with random weight initialization CNNs. Because of this, we find that reporting association results for suboptimal solutions could be misleading to the reader and therefore we have removed Table 14 in Appendix G and any text in Section 2.5 referring to this table.

The authors mentioned that PAD was estimated to be heritable in prior studies. It would be helpful to compute the heritability of PAD in the UK Biobank as a comparison.

- a. We have added a SNP-heritability estimate of PAD in the UK Biobank to Section 2.5:

"Running LD score regression [2] on the GWAS results, we estimated the SNP-heritability

³https://scikit-learn.org/stable/modules/generated/sklearn.metrics.r2_score.html

⁴<https://stats.stackexchange.com/questions/12900/when-is-r-squared-negative>

for PAD to be $h_{snp}^2 = 0.264$ (95% CI = [0.178, 0.350])⁵. This h_{snp}^2 estimate is close to the one previously estimated by Kaufmann et al. [8] [$h_{snp}^2 = 0.1828$ ($SE = 0.02$)]. And predictably our h_{snp}^2 is lower than the narrow-sense heritability estimate of PAD estimated by Cole et al. [6] [$h^2 = 0.66$ ($SE = 0.09$)] using a twin study sample."

And a new section to methods (Section 4.6) describing details of how the heritability analysis was performed:

"To estimate SNP-heritability (h_{snp}^2) we ran LD score regression [2] on the PAD GWAS summary statistics. We used the ldsc command line tool⁶ and followed standard procedure when running it. We used precomputed European 1000 Genomes phase 3 LD Scores, and filtered out rare variants with MAF < 0.01 and imputation quality score < 0.9 before training the LD score regression model. If the slope of this regression model is adjusted for the sample size and the number of SNPs it can be used to calculate an estimate of h_{snp}^2 [2]. Additionally, the intercept of the model minus one is a measure of confounding bias in the test statistics, due to confounding effect such as cryptic relatedness and population stratification [2]."

Computational cost (time and memory use) of CNN should be reported.

- a. A sentence reporting the computational cost of each CNN has been added to Section 4.3.

"Our CNN implementation uses about ~8 GB of memory and the training time using an Intel Xeon Gold 6130 Processor CPU with 32GB of RAM, and an NVIDIA Tesla V100 16GB GPU was about two days."

The flow of the text can be further improved. Currently one needs to go back and forth between the results and methods to figure out what has been done.

- a. The structure of the papers follows the guidelines set by *Nature Communications* (results section before methods). The way the paper is currently written is that the main text includes a general outline of the experiments and the more technical details are left for the methods section. The benefit of the current format is that it simplifies the reading of the paper for those with experience with the technical aspects of the paper, such as deep learning. As things stand, we are not prepared to change the flow of the manuscript unless the reviewer specifies what information he feels is lacking from the results section.

Page 9: "We also see that the other three sequence variants and PAD are associated with numerous structural brain phenotypes"

This is sort of circular because search of the three sequence variants were restricted to those associated with brain phenotypes.

- a. The sentence in consideration has been changed to reflect this (Section 2.5):

"Although it was known a priori that the other three sequence variants would associate with structural brain phenotypes, the specific structural brain phenotypes that associate with these variants are listed in Tables 10-12."

Page 9: "we scanned for the phenotype effects of all five single-nucleotide polymorphisms (SNPs) in the UK Biobank data analyzed by the Roslin Institute" The main text should be more self-contained even if detailed results are presented in the supplementary.

What are those phenotypes? Is this a PheWAS study? What is the main finding/conclusion?

- a. The Roslin gene ATLAS is a large database of associations between hundreds of traits and millions of variants using the UK Biobank cohort [3]. Originally, the phenotype associations from the Roslin

⁵Additionally, the intercept of the LD score regression model is equal to 0.991 (95% CI = [0.979, 1.003]), which indicates that the model did not find any evidence of confounding effects in the PAD GWAS results.

⁶<https://github.com/bulik/ldsc>

gene ATLAS were included for the variants reported in our manuscript to show that the PAD variants associate with more phenotypes than we had access to in our study. These results are not important to our main findings. Therefore, we decided to remove them from the new version of the manuscript.

References

- [1] Christopher M Bishop. *Pattern recognition and machine learning*. springer, 2006.
- [2] Brendan K Bulik-Sullivan, Po-Ru Loh, Hilary K Finucane, Stephan Ripke, Jian Yang, Nick Patterson, Mark J Daly, Alkes L Price, Benjamin M Neale, Schizophrenia Working Group of the Psychiatric Genomics Consortium, et al. Ld score regression distinguishes confounding from polygenicity in genome-wide association studies. *Nature genetics*, 47(3):291, 2015.
- [3] Oriol Canela-Xandri, Konrad Rawlik, and Albert Tenesa. An atlas of genetic associations in uk biobank. *Nature genetics*, 50(11):1593, 2018.
- [4] François Chollet et al. Keras. <https://keras.io>, 2015.
- [5] James H Cole, Robert Leech, David J Sharp, and Alzheimer’s Disease Neuroimaging Initiative. Prediction of brain age suggests accelerated atrophy after traumatic brain injury. *Annals of neurology*, 77(4):571–581, 2015.
- [6] James H Cole, Rudra PK Poudel, Dimosthenis Tsagkrasoulis, Matthan WA Caan, Claire Steves, Tim D Spector, and Giovanni Montana. Predicting brain age with deep learning from raw imaging data results in a reliable and heritable biomarker. *NeuroImage*, 163:115–124, 2017.
- [7] Katja Franke, Gabriel Ziegler, Stefan Klöppel, Christian Gaser, Alzheimer’s Disease Neuroimaging Initiative, et al. Estimating the age of healthy subjects from t1-weighted mri scans using kernel methods: exploring the influence of various parameters. *Neuroimage*, 50(3):883–892, 2010.
- [8] Tobias Kaufmann, Dennis van der Meer, Nhat Trung Doan, Emanuel Schwarz, Martina J. Lund, Ingrid Agartz, Dag Alnæs, Deanna M. Barch, Ramona Baur-Streubel, Alessandro Bertolino, Francesco Bettella, Mona K. Beyer, Erlend Bøen, Stefan Borgwardt, Christine L. Brandt, Jan Buitelaar, Elisabeth G. Celius, Simon Cervenka, Annette Conzelmann, Aldo Córdova-Palomera, Anders M. Dale, Dominique J.-F. de Quervain, Pasquale Di Carlo, Srdjan Djurovic, Erlend S. Dørum, Sarah Eisenacher, Torbjørn Elvsashagen, Thomas Espeseth, Helena Fatouros-Bergman, Lena Flyckt, Barbara Franke, Oleksandr Frei, Beathe Haatveit, Asta K. Haberg, Hanne F. Harbo, Catharina A. Hartman, Dirk Heslenfeld, Pieter J. Hoekstra, Einar A. Høgestøl, Terry Jernigan, Rune Jonassen, and Jönsson. Genetics of brain age suggest an overlap with common brain disorders. *bioRxiv*, 2018. doi: 10.1101/303164. URL <https://www.biorxiv.org/content/early/2018/04/17/303164>.
- [9] Hyunwoo Lee, Kunio Nakamura, Sridar Narayanan, Robert A Brown, Douglas L Arnold, Alzheimer’s Disease Neuroimaging Initiative, et al. Estimating and accounting for the effect of mri scanner changes on longitudinal whole-brain volume change measurements. *NeuroImage*, 184:555–565, 2019.
- [10] Amber NV Ruigrok, Gholamreza Salimi-Khorshidi, Meng-Chuan Lai, Simon Baron-Cohen, Michael V Lombardo, Roger J Tait, and John Suckling. A meta-analysis of sex differences in human brain structure. *Neuroscience & Biobehavioral Reviews*, 39:34–50, 2014.
- [11] Pierrick Wainschtein, Deepti P Jain, Loic Yengo, Zhili Zheng, L Adrienne Cupples, Aladdin H Shadyab, Barbara McKnight, Benjamin M Shoemaker, Braxton D Mitchell, Bruce M Psaty, et al. Recovery of trait heritability from whole genome sequence data. *bioRxiv*, page 588020, 2019.

REVIEWERS' COMMENTS:

Reviewer #2 (Remarks to the Author):

The authors have done a good job responding to the reviewers comments and the manuscript is further improved.

I'm still don't think that the authors adequately justify why sex and scanner should be used in an age prediction model. My understanding is that covariates should be included if there are likely to relate to the outcome measure (i.e., age), not other predictor variables (i.e., brain imaging measures). If they did want to justify the value of adding these measures, then it would be possible to do so using sensitivity analyses (e.g., with/without covariates), applied to males/females only etc., but I believe that this is beyond the scope of the current paper and perhaps focus for another occasion.